# Cryo-EM structure of the human ferritin–transferrin receptor 1 complex

Linda Celeste Montemiglio[1,2,3], Claudia Testi[1,4], Pierpaolo Ceci [2], Elisabetta Falvo[2], Martina Pitea[1], Carmelinda Savino[2], Alessandro Arcovito [5,6], Giovanna Peruzzi [4], Paola Baiocco[4], Filippo Mancia[7], Alberto Boffi[1], Amédée des Georges [8,9,10] & Beatrice Vallone [1,3]

Human transferrin receptor 1 (CD71) guarantees iron supply by endocytosis upon binding of iron-loaded transferrin and ferritin. Arenaviruses and the malaria parasite exploit CD71 for cell invasion and epitopes on CD71 for interaction with transferrin and pathogenic hosts were identified. Here, we provide the molecular basis of the CD71 ectodomain-human ferritin interaction by determining the 3.9 Å resolution single-particle cryo-electron microscopy structure of their complex and by validating our structural findings in a cellular context. The contact surfaces between the heavy-chain ferritin and CD71 largely overlap with arenaviruses and *Plasmodium vivax* binding regions in the apical part of the receptor ectodomain. Our data account for transferrin-independent binding of ferritin to CD71 and suggest that select pathogens may have adapted to enter cells by mimicking the ferritin access gate.

[1] Department of Biochemical Sciences "Alessandro Rossi Fanelli", Sapienza University of Rome, P.le A. Moro 5, 00185 Rome, Italy. [2] Institute of Molecular Biology and Pathology, National Research Council, P.le A. Moro 5, 00185 Rome, Italy. [3] Istituto Pasteur-Fondazione Cenci Bolognetti, Dipartimento di Scienze Biochimiche "A. Rossi Fanelli", Sapienza Università di Roma, P.le A. Moro 5, 00185 Rome, Italy. [4] Center for Life Nano Science @ Sapienza, Istituto Italiano di Tecnologia, V.le Regina Elena 291, 00161 Rome, Italy. [5] Istituto di Biochimica e Biochimica Clinica, Università Cattolica del Sacro Cuore, Largo F. Vito 1, 00168 Rome, Italy. [6] Fondazione Policlinico Universitario Agostino Gemelli-IRCCS, Largo F. Vito 1, 00168 Rome, Italy. [7] Department of Physiology and Cellular Biophysics, Russ Berrie Pavilion, Columbia University Medical Center, 1150 St Nicholas Ave, New York, NY 10032, USA. [8] Advanced Science Research Center at The Graduate Center of the City University of New York, 85 Saint Nicholas Terrace, New York, NY 10031, USA. [9] Department of Chemistry and Biochemistry, City College of New York, New York, NY 10031, USA. [10] Programs in Biochemistry and Chemistry, The Graduate Center of the City University of New York, New York, NY 10016, USA. These authors contributed equally: Linda Celeste Montemiglio, Claudia Testi. Correspondence and requests for materials should be addressed to A.d.G. (email: amedee.desgeorges@asrc.cuny.edu) or to B.V. (email: beatrice.vallone@uniroma1.it)

Human transferrin receptor 1 (CD71 or hTfR1) is a promiscuous and ubiquitously expressed cell entry carrier whose primary function is to import iron in response to variations in intracellular concentration of this essential element. Iron uptake is mediated by the internalization of the transferrin (Tf)–iron complex through receptor-mediated constitutive endocytosis via a clathrin-dependent pathway[1]. Once the iron cargo is delivered, the receptor is recycled back to the cell surface and apo-Tf is released into the bloodstream[2]. CD71 has been also shown to mediate the uptake of heavy-chain ferritin (H-Ft) from serum as an alternative or additional source of bioavailable iron[3]. CD71 is also a preferred entry carrier for human pathogenic arenaviruses[4–8] and hepatitis C virus[9], as well as feline-specific and canine-specific parvoviruses[10]. Viral systems recognize epitopes on the host-encoded CD71 receptor through their surface spike glycoproteins, allowing the internalization of the complex. Recently, *Plasmodium vivax*, the most common malaria parasite, was demonstrated to access reticulocyte cytoplasm by recognizing the same CD71 receptor epitope as arenaviruses[11,12].

CD71 is a homodimeric type II transmembrane protein composed of a small cytoplasmic domain, a single-pass transmembrane region, and a complex extracellular domain. Each monomer of the ectodomain is subdivided in a protease-like domain in contact with the cell membrane, a helical domain comprising the dimer contact regions, and an apical domain (Fig. 1a)[13]. The ectodomain displays ligand-binding sites for diverse proteins: its basal portion (formed by the protease-like and the helical domains) binds Tf[14–16] and the dimer interface region forms a complex with the hereditary hemochromatosis factor (HFE)[17] (Fig. 1b). The upper part of the apical domain has been shown to interact with arenaviruses[4,8] and with the *P. vivax* invasion protein *Pv*RBP2b[12] (Fig. 1b).

In this framework, a key missing piece of information concerns the structural basis of the interaction between CD71 and H-Ft. Experimental evidence was provided for a scarce competition between ferritin and Tf for CD71 binding, thus pointing out the possibility of the existence of different epitopes for the two protein ligands[3,18]. Recently, an exposed loop region in the H-Ft subunit was identified that, transplanted in an archaeal ferritin, originally unable to recognize the human CD71 receptor, was sufficient to induce binding of this chimeric protein to the receptor[19].

The importance of the CD71/H-Ft interaction is dictated by the emerging physiological and pathological significance of the circulating ferritin and its scavenger receptor[20,21]. Moreover, nano-sized H-Ft homopolymers have moved to the center stage of nanomedicine research as theranostic agents[22], due to their unique cargo capabilities for small therapeutic molecules or isotopic tracers coupled to selectivity towards CD71[23–25]. CD71 is highly expressed in the most common cancer cell types, further highlighting the interest for this receptor as a privileged target for the selective delivery of cytotoxic drugs coupled to Tf, ferritin, or monoclonal antibody drug conjugates[26–28].

We used single-particle cryo-electron microscopy to solve the structure of H-chain ferritin bound to human CD71 ectodomain to 3.9 Å resolution, unveiling the structural determinants that govern their recognition.

## Results and discussion

**H-Ft binds the CD71 receptor in a virus-like fashion.** H-Ft binds the CD71 receptor through four specific contact regions on the apical domain, covering an overall area of ~1900 Å² (Fig. 2, Supplementary Figures 1–3 and Supplementary Table 1). As depicted in Fig. 2, the four contact regions comprise: (i) a motif of six amino acids, from R208 to L212 and N215 on the βII-2 strand; (ii) residues E343, K344, and N348 on the αII-2 helix. We refer to these residues as "common contacts" on CD71, since they represent the key structural determinants for binding of arenavirus (MACV) and plasmodial *Pv*RBP2b proteins. Additionally,

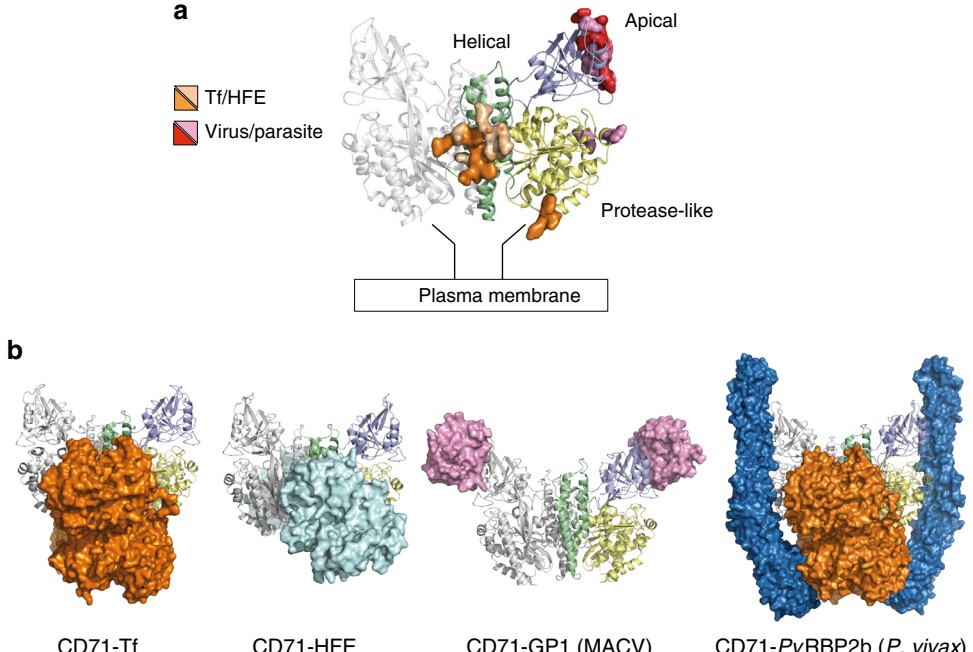

**Fig. 1** CD71 receptor: ligand recognition epitopes and binding modes. CD71 homodimer is shown in ribbon representation (pdb 3KAS[8]). One monomer is in light gray, the other is colored variably to the receptor domains (apical, light blue; protease-like, yellow; helical, green). **a** CD71 residues identified as recognition epitopes for Tf/HFE and viruses/parasite are represented as orange/wheat and red/pink surfaces, respectively. **b** CD71 receptor is shown bound to Tf (orange surface, pdb 1SUV[14]), HFE (cyan surface, pdb 1DE4[17]), GP1 protein of MACV (pink surface, pdb 3KAS), and Tf (orange surface) and *Pv*RBP2b from *P. vivax* (blue surface, pdb 6D04[12]).

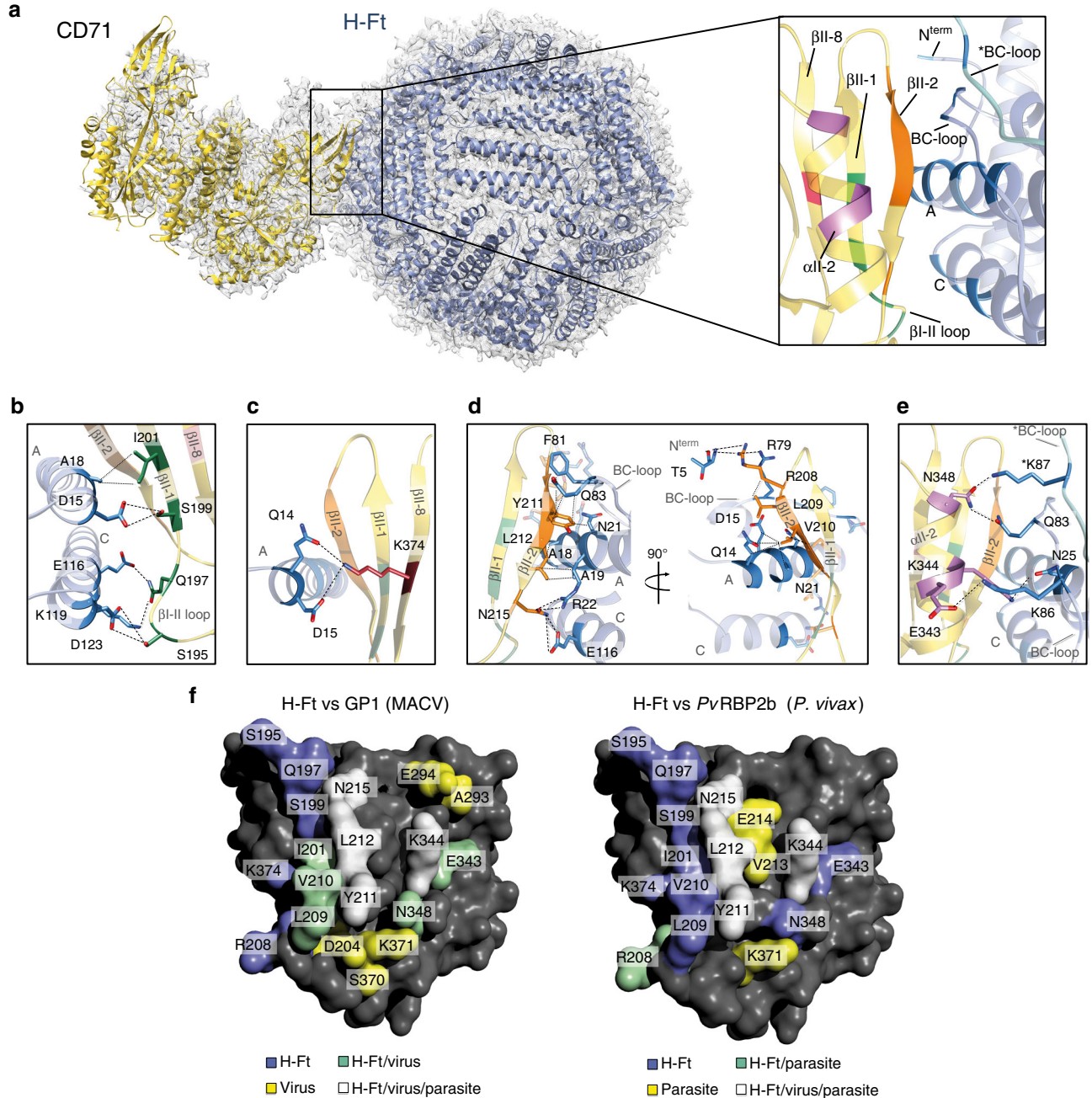

**Fig. 2** Structure of the human CD71/H-Ft complex. **a** Atomic model of the complex of human CD71 receptor (yellow) and human H-Ft (light blue) fitted in the cryo-EM map at a global resolution of 3.9 Å (gray mesh) is shown. On the side, close-up view of the contact region. The interacting residues of CD71 are highlighted by different colors depending on the specific secondary structural element of the apical domain to which they belong. Interacting residues of H-Ft are colored in light blue. **b–e** CD71 and H-Ft interacting regions. Panels **b** and **c** show the "exclusive contacts"; panels **d** and **e** show the "common contacts". Contacting residues within 5 Å distance are shown in sticks, labeled and colored according to the color code used for secondary structure in panel **a** (right side). Dashed black lines indicate electrostatic interactions. Dotted gray lines represent hydrophobic contacts. **f** Surface representation of the human CD71 apical domain (dark gray). Ligand specific and overlapping residues contacted by the human H-Ft, GP1 of Machupo virus[8] and *Pv*RBP2b of *P. vivax*[12] are mapped out following the color code shown at the bottom

"exclusive contact" regions on CD71 for H-Ft comprise (iii) the βI-1–βII-1 loop and the βII-1 strand (S195, E197, S199, and I202) and (iv) the βII-8 strand (K374). The H-Ft-binding counterpart regions are: (i) the external BC loop (R79, F81, Q83, K86, K87); (ii) the N-terminus of the A-helix (Q14, D15, E17-A19, N21, R22, N25); (iii) the C-terminus of the C helix (E116, K119, D123) (Fig. 2).

The H-Ft "exclusive contacts" pairwise interactions are shown in Fig. 2b and c, where: (i) CD71 βII-1 strand contacts H-Ft via

the A helix; (ii) βI-1–βII-1 loop interacts with the H-Ft C helix; (iii) K374 on βII-8 of CD71 electrostatically interacts with Q14 at the beginning of H-Ft A helix.

The "common contacts" with CD71, shared between pathogens and H-Ft, are shown in Fig. 2d and are established by the βII-2 strand of CD71 with the H-Ft A helix, BC loop, and C helix, and with T5 at the H-Ft N-Terminus. Further common interactions are established between the CD71 αII-2 and H-Ft A helix, the BC loop and K87 on the BC loop of a flanking H-Ft monomer

(Fig. 2e; the detailed contacts are given in Supplementary Table 2).

Notably, the amino acids involved in "common contacts" coincide with amino acids on CD71 that lead to gain or loss of interaction with pathogen-binding proteins upon mutations[12,29] (Fig. 2f and Supplementary Table 3). These cluster in the βII-2 and αII-2 regions, which appear to be essential on CD71 for binding with various partners. Therefore, in order to identify the determinants on H-Ft for binding to CD71 we selected and mutated H-Ft residues involved in "common contacts" which are not conserved in ferritins shown to be unable to bind human CD71, i.e. human Light-chain ferritin[3] (Hum L-Ft or L-Ft) and *Archeoglobus fulgidus* ferritin[19] (AfFt) (Supplementary Figure 4).

**Mutations at common contacts tune ferritin–CD71 interaction.** We produced three multiple mutants of residues peculiar of human H-Ft: (i) mutant A lacking the polar residues at the N-terminal of the A helix (Q14A/D15A/R22A), (ii) mutant B lacking F81 and Q83 on the external BC-loop (F81A/Q83A), and (iii) mutant C that combines A and B mutations (Q14A/D15A/R22A/F81A/Q83A) (Supplementary Figure 4). Surface plasmon resonance (SPR) measurements using wild-type or mutant H-Fts as analytes and CD71 as ligand showed that the binding affinity for the receptor is reduced of about two orders of magnitude in mutants A and B as compared to the wild type, and abolished in mutant C (Fig. 3a, Supplementary Tables 4 and 5). In particular a critical drop of the $k_{on}$ value is increased across mutants B and C, suggesting that mutations at the BC loop have a dominant role in

impairing the CD71/H–Ft interaction, likely due to the loss of contact of F81 and Q83 with Y211 on CD71 βII-2 and Q83 with N348 on CD71 αII-2. Consistently, fluorescence-activated cell sorting (FACS) (Fig. 3b, Supplementary Figure 5) and confocal microscopy (Fig. 3c) measurements on HeLa cells uptake of the three H-Ft mutants show reduced (mutant A) or negligible (mutants B and C) internalization with respect to the wild type.

All together, these results assess the critical relevance of ferritin external BC-loop in CD71 binding, recognizing the "common contact" residues Y211 and N348, crucial for binding of viruses and plasmodial proteins, whose mutations severely hamper binding and suppress internalization. However, they also highlight the relevance of Q14, D15, and R22 at the N-terminus of the A helix.

Therefore, the significant binding capability of the humanized *A. fulgidus* ferritin (Hum-AfFt) where the human H-Ft BC loop had been transplanted, is likely due not only to the presence of F81 and Q83 on the loop, but also to R22 on the A helix, serendipitously present in this archaea ferritin.

Notably, Hum L-Ft, which is unable to bind CD71[3], presents differences in seven positions (S5, T14, S22, L81, D116, A119, A123) over the total contacts required for the recognition of the receptor (Supplementary Figure 4a). We found that L-Ft to H-Ft mutations at these positions (mutant D) confer binding capability to CD71 with an affinity similar to the one observed for H-Ft (Supplementary Figure 6).

**Significance of CD71 apical domain.** In this work, we identified the specific sites on CD71 to be hooked by H-ferritin for

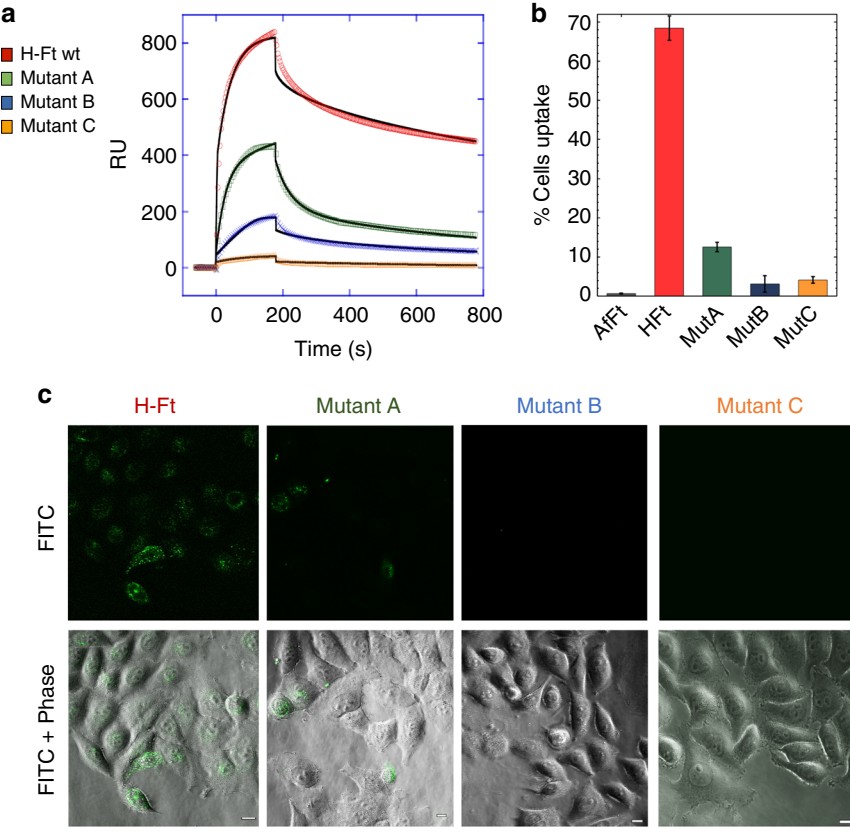

**Fig. 3** Characterization of human H-Ft wild type and mutants binding to CD71. **a** SPR sensograms of the interaction between the immobilized his-tagged CD71 receptor and H-Fts (wild-type and mutants), used as analytes. Fits are reported as black lines. Source data are provided as a Source Data file. **b** Ferritins uptake in HeLa cells has been quantified by flow cytometry. The percentage of cells internalizing AfFt-FITC (here used as negative control[19]), H-Ft-FITC, MutA-FITC, MutB-FITC, and MutC-FITC is shown as mean ± s.e.m. for $n = 3$ independent experiments. Source data are provided as a Source Data file. **c** Internalization of ferritins in HeLa cells observed at a ×60 confocal microscope, here shown as single FITC and overlay images with phase-contrast. Scale bars: 10 µm

physiological access to cell through the CD71 "iron door". Our results show that even a single subunit of H-chain over 24 can be recognized by CD71 and this might be sufficient to favor the cellular internalization of the mixed H/L-chain 24-mer ferritin through this route of access, even though in serum the L-chain is prevalent[30]. Moreover, we provide structural evidence that Tf and ferritin exploit alternative epitopes on the same receptor, allowing differential regulation of iron, as in the case of the HFE[3,17].

Importantly, pathogens have adapted to exploit the ferritin epitope to unlock cellular barriers by mimicking this physiological interaction with the CD71 apical domain. In this framework, changes due to single nucleotide polymorphisms (SNPs) within the CD71 apical domain may account for permissivity of virus or parasite entry, but cannot be considered neutral with respect to ferritin uptake. Along this line, CD71 species-specific variants must conveniently be matched to H-Ft co-evolved variants in order to conserve the physiological functions of serum circulating ferritin.

In conclusion, our work provides a sound structural basis to elaborate on the possibility of developing alternative ferritin-like anti-viral or anti-parasite therapeutic ligand, be it an antibody or a peptidomimetic capable of blocking the "common contacts" epitope on CD71 residue, and to further engineering ferritins as theranostic agents.

## Methods

**Cloning expression and purification of human H-Ft and CD71.** The genes encoding the human apo-H-chain ferritin (H-Ft), mutants A (Q14A/D15A/R22A), B (F81A/Q83A), human apo-L-chain ferritin (Hum L-Ft or L-Ft) and its mutant D (S5T/T14Q/S22R/L81F/D116E/A119K/A123D following H-Ft numbering), and *Archaeoglobus fulgidus* ferritin (AfFt) were designed, synthesized, optimized for *Escherichia coli* codon usage and cloned in pET11b vector by Geneart (Geneart AG) (Supplementary Table 6). Mutant C was produced by using the Quick Change Lightning Kit (Agilent Technologies, Santa Clara, CA, USA) according to the manufacturer's instructions and using mutant A as template (Supplementary Table 6). All apo-H-Ft variants, L-Ft expression, and purification were performed as reported by Falvo et al.[27], yielding about 90 and 40 mg of pure protein per litre of bacterial culture, respectively, for the H-ferritins and for the L-ferritin. Briefly, *E. coli* BL21 (DE3) cells harboring recombinant plasmids were grown to $OD_{600}$ 0.6 at 37 °C in 1 L of ampicillin-containing Terrific Broth (TB) medium. Gene expression was induced by addition of 0.5 mM isopropyl-1-thio-ß-D-galactopyranoside (IPTG) and cells were further grown at 22 °C overnight. After cell harvesting, pellet was suspended in 50 mM Tris–HCl, 0.5 mM dithiothreitol (DTT), 1 mM ethylenediamine tetra-acetic acid (EDTA), and 300 mM NaCl, pH 7.5, and disrupted by sonication in the presence of 1 mM phenylmethylsulfonyl fluoride (PMFS). The lysate was centrifuged and the supernatant containing the soluble fraction was treated 40 min at 37 °C with 0.1 mg/mL DNase supplied with 10 mM $MgCl_2$, heated to 55 °C for 8 min, and then centrifuged to remove denatured proteins. The recovered supernatant was heated a second time to 72 °C for 8 min, and then centrifuged. The recovered supernatant was precipitated in 75% ammonium sulfate. The pellet was resuspended and dialyzed overnight against phosphate-buffered saline (PBS) pH 7.5 and then loaded onto a HiTrap Q HP column (Q Sepharose High Performance GE Healthcare, Boston, USA). The recovered samples were ultracentrifuged at 35,000 rpm for 55 min at 6 °C using a Beckman L8-70M ultracentrifuge (Beckman Coultier). The recovered supernatant was then precipitated using ammonium sulfate at 65% saturation. The pellet was resuspended and dialyzed overnight against PBS pH 7.5, pooled, concentrated, and sterile-filtered before storing at 4 °C. The L-Ft mutant D had low purification yield due to a high fraction of expressed protein being lost in the inclusion bodies. Purity was assessed by SDS–PAGE; protein concentration was determined spectrophotometrically at 280 nm using a molar extinction coefficient (on a 24-mer basis) of $4.56*10^5 M^{-1} cm^{-1}$ (ProtParam sofware, http://www.expasy.org). In order to assess the 24-mer assembly of all the produced ferritin variants, size exclusion chromatography (SEC)-HPLC experiments were performed on samples at 2 mg/mL concentration, using a Superose 6 gel-filtration column equilibrated with PBS at pH 7.4 (Supplementary Figure 4b). Traces were analyzed with Origin 8.0 (Originlab Corporation, Northampton, MA).

AfFt expression was performed in *E. coli* BL21 (DE3) and induced with 1 mM IPTG at $OD_{600} = 0.6$. After 3 h at 37 °C cells were harvested by centrifugation. Pellet was resuspended in a lysis buffer containing 20 mM HEPES, pH 7.5, 200 mM NaCl, 1 mM TCEP (tris(2-carboxiethyl)phosphine), and a cOmplete™ Mini Protease Inhibitor Cocktail Tablet (Roche). Cells were disrupted by sonication and the soluble fraction was purified by heat treatment at 80 °C for 10 min, followed by centrifugation (15,000 rpm at 4 °C for 1 h). The soluble protein was further purified by ammonium sulfate precipitation. The precipitated fraction at 70% ammonium sulfate was

resuspended in 20 mM HEPES, 50 mM $MgCl_2$, pH 7.5 and dialyzed versus the same buffer. SEC was then performed using a HiLoad 26/600 Superdex 200 GL column previously equilibrated in the same buffer. Sample purity was estimated by SDS–PAGE and protein concentration was calculated spectrophotometrically using an extinction coefficient at 280 nm of 33,900 $M^{-1} cm^{-1}$. Protein yield was ~40 mg $L^{-1}$ culture[31].

The gene encoding the ectodomain of human CD71 (residues 121–760) was extracted by pcr from the plasmid pAcGP67A-TfR[32] (Addgene, Cambridge, MA) and BamHI/EcoRI inserted using the Gibson cloning method and fused at the 3′ of the Kozak sequence of the pα−H mammalian expression vector modified by the addition of the hydrophobic leader peptide from the baculovirus protein gp67 (pα−H BiP). An octa-histidine tag was also placed at the C-terminus of the CD71 gene. The expression plasmid pα−H BiP/TfR1 was transiently transfected in HEK 293 using polyethylenimine (PEI) as transfection agent. Cells were grown in FreeStyle 293 expression media (ThermoFisher Scientific, Hampton, USA) supplemented with 1% of fetal bovine serum (FBS) at 37 °C in a humidified atmosphere of 5% $CO_2$. After 96 h, cells were harvested and CD71 was purified from supernatants using Ni-affinity or Co-affinity chromatography. Supernatant was filtered and incubated with the resin after addition of 50 mM sodium phosphate buffer, pH 8.0, together with 200 mM NaCl and 20 mM imidazole; 250 mM imidazole was used to elute CD71. The protein sample was stored at −80 °C in 50 mM sodium phosphate, 200 mM NaCl, pH 8. Quality and quantity of purified protein was evaluated by SDS–PAGE and UV/visible spectra using the theoretical $\varepsilon_{280 nm}$ 96,260 $M^{-1} cm^{-1}$.

**CD71/H-Ft complex preparation.** In vitro, incubation of H-Ft and CD71 at different stoichiometric ratios results in protein aggregation due to the presence of multiple binding sites on both H-Ft and CD71, forming insoluble precipitates. We managed to isolate a soluble sample of CD71/H-Ft complex by means of a pull-down experiment[3]. Two hundred and fifty micrograms of 8xHis-tag-fused CD71 was incubated with 100 μL of TALON resin (TALON Superflow Metal Affinity Resin, Ge Healthcare, UK) in 25 mM Tris–HCl, 150 mM NaCl, 1% PEG 8000, and 10 mM Imidazole, pH 7.2 (buffer A), for 60 min at 4 °C, under rotation. After several washes with buffer A, CD71-conjugated beads were incubated with 950 μg of H-Ft for 90 min at 4 °C under rotation. The beads were washed increasing imidazole concentration in buffer A up to 30 mM, and the complex was eluted in 350 μL using 290 mM imidazole in buffer A. As a control experiment, CD71-unconjugated beads were also incubated with 950 μg of H-Ft for 90 min, following the same procedure adopted to isolate the complex. The pull-down assay final samples were analyzed by SDS/PAGE (Supplementary Figure 1a).

**Grids preparation for cryo-electron microscopy.** The CD71/H-Ft complex eluted from Talon resin at a concentration of 0.2 mg/mL in 25 mM Tris–HCl, 150 mM NaCl, 1% PEG 8000, and 290 mM Imidazole, pH 7.2 was used for grid preparation immediately after the pull-down experiment, without buffer exchange and sample concentration. Two datasets were collected using the same batch of grids to obtain the final map.

Holey-gold R0.9/1 (dataset 1) and holey-carbon R1.2/1.3 (dataset 2) grids (Quantifoil Micro Tools GmbH) covered by 2 nm film of carbon were prepared as described[33]. Grid surfaces were treated with plasma cleaning using a mixture of Ar and $O_2$ for 60 s before applying 3 μL of sample. The screening of several blotting conditions revealed that the time between sample application to the grid and plunge into ethane affects the number of particles per field and their distribution. The dataset 1 grid was prepared with a 135 s wait time and the dataset 2 grid with a 90 s wait time. Grids were then blotted for 1 s (100% humidity, 4 °C, force 4) with filter paper and vitrified by rapidly plunging into liquid ethane at −180 °C using a Vitrobot Mark IV (FEI, Hillsboro)[34].

**Data collection.** The first dataset (dataset 1, 690 micrographs, Supplementary Figure 3) was imaged using a FEI Titan Halo (ThermoFisher Scientific, Eindhoven) operating at 300 kV acceleration voltage, while the specimen was maintained at liquid nitrogen temperature using a Gatan 626 side entry cryo-holder (Gatan, Pleasanton). Images were recorded using the automated acquisition program Leginon[35]. We used a Gatan K2 Summit direct-detector camera (Gatan, Pleasanton) operating in counting mode, with a calibrated pixel size of 1.15 Å on the object scale. Images were typically recorded with a defocus range between −0.7 and −3.0 μm. Movies were acquired with a total exposure time of 12 s (60 frames/image), with exposure rate of 7.8 electrons/pix/s.

A second dataset (dataset 2, 573 micrographs, Supplementary Figure 3) was imaged using a 300 kV Titan Krios (ThermoFisher Scientific, Eindhoven). The dataset was collected automatically using EPU (ThermoFisher Scientific, Eindhoven). Images were recorded on a Gatan K2 Summit direct-detector camera (Gatan, Pleasanton) equipped with a Gatan Bioquantum LS/967 energy filter and operating in super-resolution mode, using a calibrated pixel size of 1.33 Å on the object scale. Images were typically recorded with a defocus range between −1.0 and −3.0 μm. Movies were acquired in electron counting mode, the total exposure time was set to 12 s (40 frames/image), with exposure rate of 6.2 e⁻/pix/s.

**Image processing.** The main steps of data analysis are schematized in the workflow shown in Supplementary Figure 3.

Micrograph frames collected in both datasets (Supplementary Figure 1b) were aligned for beam-induced motion correction and drift with MotionCor2[36]; global contrast transfer function was calculated using Gctf[37]. Micrographs with resolution limits ≤6 Å (dataset 1) or ≤5 Å (dataset 2) were kept for further processing.

All subsequent data analysis was carried out using RELION 2.0[38]. More than 1100 particles for both datasets were manually picked to produce a reference for the automated particle picking procedure implemented in RELION[39]. A total of 140,567 and 194,501 particles were automatically picked, respectively, from dataset 1 and dataset 2 and extracted from the original micrographs. After the extraction, particles were initially classified in 2D using $K = 100$ classes. A second round of 2D classification was performed on each dataset with $K = 25$ classes. The best 2D class averages clearly showed the CD71/H-Ft complex in different orientations, revealing some extent of sample heterogeneity due to the presence of alternative populations endowed with different stoichiometry (Supplementary Figure 1c). A set of 27,690 particles, belonging to good 2D classes of dataset 1, was selected and subjected to a first round of 3D classification, using eight classes and without imposing any symmetry; the cryo-EM apo-ferritin map (EMDB code EMD-2788[33]) was used as reference model, filtered at 60 Å. This procedure gave only one class (8860 particles) where a 1:1 = H-Ft:CD71 complex was clearly displayed. The resulting map was used as reference (filtered at 60 Å) to run a second round of 3D classification, which resulted in two better-resolved classes (total 17,370 particles). These classes (see Supplementary Figure 3) were selected for further refinement using the 3D Autorefine procedure, applying a spherical mask of 290 Å diameter. This resulted in a map at 8.2 Å resolution.

The reconstructed map of dataset 1 (filtered at 40 Å) was used as reference model for the 3D classification of particles selected from 2D classification of dataset 2 (73,700 particles). Good classes (Supplementary Figure 3) were selected for further refinement using the 3D Autorefine procedure (25,870 particles, spherical mask of 290 Å diameter), which yielded a map at 5.5 Å resolution.

With the aim to further increase the resolution of the contacting region between the two molecules, particles used for the 3D refinement of both datasets were joined (total 43,240 particles) and subjected to one round of 3D refinement, imposing O symmetry (i.e., ferritin point-group symmetry) and applying a smaller mask diameter of 180 Å, to only include the ferritin molecule. This yielded a 4.8 Å map of ferritin re-centered to its center of symmetry (Supplementary Figure 3). We used the corresponding particle alignment parameters to perform symmetry expansion of this dataset using RELION (relion_particle_symmetry_expand). This allowed us to subdivide ferritin particles in individual subunits in order to identify by classification all ferritin subunits bound to a Tf receptor[40]. The dataset was thus artificially expanded according to the pseudo-symmetric O point group and enlarged 24-fold, resulting in 1,037,760 particles, that were 3D Classified using C1 symmetry, no image alignment and a mask generated from the complex with a single receptor bound. This classification round identified all subunits bearing a receptor bound. All four classes generated were subjected to a second round of 3D classification, this time with local image alignment, and only two of them allowed the identification of subclasses of ferritin bound to the receptor. Therefore, three H-Ft bound subclasses were combined (total 53,878 particles) and refined. After postprocessing, the resolution measured using a mask (9 pixel extension, 24 pixel soft edge) only including ferritin and the receptor contact sites was estimated to be 4.4 Å resolution (Supplementary Figure 2d).

The refined particles were exported into cisTEM to perform per-particle defocus refinement procedure implemented in cisTEM[41]. The resulting final map resolution of H-Ft and CD71 was improved up to 3.9 Å, clearly showing secondary structures and bulky side chains of interacting residues, which allowed more precise model building. Statistic information of the 5.5 Å and of the 3.9 Å density maps is reported in Supplementary Table 1.

**Resolution estimation**. The overall resolution of the two maps at 5.5 and 4.4 Å obtained with RELION and the one at 3.9 Å obtained with cisTEM (Supplementary Figure 2a, c and e, f) was estimated with the Fourier shell correlation (FSC) = 0.143 criterion, based on the 'gold-standard' protocol[42] using a mask around the complex density. To estimate the resolution at the CD71/H-Ft interacting region we applied a spherical mask, created with Chimera[43], only including this portion. The resolution is 3.9 Å based on the gold-standard 0.143 FSC criterion (Supplementary Figure 2g). The input maps were corrected for the modulation transfer function (MTF) of K2 detector and sharpened using negative temperature B factors as estimated by RELION (Supplementary Table 1). Local resolution was measured for the map at 5.5 and 4.4 Å using ResMap[44] (Supplementary Figure 2b and d). The electron density maps obtained at 5.5 and at 3.9 Å resolution were sharpened using autosharpen in Phenix[45]. UCSF Chimera[43] and PyMOL (http://pymol.sourceforge.net) were used for graphical visualizations.

**Model building and refinement**. The program Chimera was initially employed to rigid body fit the crystal structures of H-Ft (PDB code 3AJO[46]) and CD71 (PDB code 3KAS[8]) into the sharpened electron density map obtained at 5.5 Å resolution, which was further refined as rigid body with Phenix real_space_refinement. In this map, both CD71 and H-Ft were visible at the level of their secondary structures. The resulting CD71/H-Ft model was then rigid body fitted into the sharpened 3.9 Å density map using UCSF Chimera. Model building of H-Ft and of the CD71 contact region was performed manually using Coot[47]. Given the high quality of the

reconstructed map, the side chains of residues exposed at the contacting interface were modeled into the electron density (Supplementary Figure 2h).

The model of the CD71/H-Ft complex was refined using the real_space_refinement routine in minimization global mode (Phenix) against the overall map at 3.9 Å, imposing secondary structure and Ramachandran restraints. Final visual inspection was performed in COOT to manually correct Ramachandran outliers. The final model was validated using MolProbity[48] and EMRinger[49] (Supplementary Table 1). All figures were produced using Pymol and UCSF Chimera.

**CD71/H-Ft and CD71/L-Ft SPR assay**. The interactions between the immobilized N-terminal His-tagged Tf receptor CD71 (ligand) and H-Ft-based constructs (analytes) were measured by SPR technique on a Biacore X100 instrument (Biacore, Uppsala, Sweden) according to the procedure previously reported [28]. Briefly, CD71 was immobilized on a Sensor Chip nitrilotriacetic acid (NTA) (GE Healthcare Europe GmbH) according to the manufacturer's instructions. The optimal experimental setup was determined and CD71 was injected at 22 μg/mL for 60 s for the multi-kinetic mode and up to 30 μg/mL in the single kinetic mode. For the first mode, analyte concentration was in the range of 1000–62.5 μg/mL. The sensor chip surface was regenerated using fresh histidine-tagged protein after every cycle of the assay. The SPR assay was performed at 25 °C, at flow rate = 30 μL/min; the association and dissociation phases were monitored for 180 and 600 s, respectively. Analytes were dissolved in degassed 10 mM PBS at pH 7.4. In this condition H-Ft retains the 24-mer assembly (Supplementary Figure 4b). To regenerate the chip, complete dissociation of the complexes was achieved by the addition of 10 mM HEPES, 150 mM NaCl, 350 mM EDTA, and 0.005% (vol/vol) surfactant P20 (pH 8.3) for 30 s before starting a new cycle. The $k_{on}$ and $k_{off}$ rates as well as the dissociation constant ($K_D$) were estimated using the Biacore X100 evaluation software according to a 1:1-binding model or alternatively a heterogeneous analyte-binding model (Supplementary Tables 4 and 5). All experimental data shown in Fig. 3a are reported at analyte concentration of 1 mg/mL. Fits are reported as black lines corresponding to a heterogeneous analyte-binding model for wild type, Mutant A and Mutant B, respectively, and to a simple 1:1 kinetic model for Mutant C.

For the single kinetic mode, analyte concentration was in the range of 300–18.75 μg/mL and the sensor chip surface was regenerated using fresh histidine-tagged protein for all ferritins tested, i.e. wild type H-Ft, L-Ft, and mutant D.

**Protein FITC labeling**. HFt, AfFt, Mutant A, Mutant B, and Mutant C were labeled with fluorescein-isothiocyanate (FITC, ThermoFisher) according to the manufacturer's standard protocol. Briefly, 2 mg/mL of the purified protein was added with 10-fold molar excess of FITC in protein storage buffer, stirring for 2 h at RT. The non-reacted dye was removed by gel filtration chromatography and the fluorescent dye to protein ratio was determined by UV spectroscopy. LC–MS spectrometry measurements of all FITC-conjugated proteins confirmed that >40% of monomers are FITC labeled.

**Cell cultures and ferritins internalization**. HeLa cells (ATCC Number: CCL-2) were grown at 37 °C in Dulbecco's modified Eagle's medium with Glutamax (DMEM, Gibco) supplemented with 10% FBS (Gibco) and 1% Penstrep (100 U/mL penicillin and 100 μg/mL streptomycin solution; Gibco). DMEM without phenol red (Sigma) supplemented with Glutamax (Invitrogen), 10% FBS and 1% Penstrep (incubation medium) was used for apo-ferritins internalization assays by FACS and confocal microscopy.

**Flow cytometry analysis**. HeLa cells were seeded on six-well plates and left 1 day prior performing FACS experiments. Upon growing medium removal and rinse with PBS, confluent cells were incubated with FITC-ferritin nanoparticles (AfFt, HFt, Mutant A, B and C as specified in each experiment) at the final concentration of 30 μg/mL for 2 h 30 min. Cells were then washed twice with PBS, detached with trypsin-EDTA (Euroclone), rinsed with PBS and resuspended in BD-FACS flow buffer. Control cells were treated in the same conditions without ferritins. Internalization of ferritins was evaluated with sample acquisition at the BD LSRFortessa (BD Biosciences, San Jose, CA, USA) equipped with a 488 nm laser and FACSDiva software (BD Biosciences version 6.1.3). Live cells were first gated by forward and side scatter area (FSC-A and SSC-A) plots, then detected in the green channel for FITC expression (530/30 nm filter) and side scatter parameter. The gate for the specific signal was set based on the control sample. Data were analyzed using FlowJo 9.3.4 software (Tree Star, Ashland, OR, USA).

**Confocal microscopy of live cells**. For apo-ferritins internalization by live imaging on a confocal microscope, HeLa cells were seeded on a μ-slide eight-well ibiTreat (ibidi) and left 1 day to grow. After the removal of the medium, cells were washed with PBS and incubated for 20 h with 30 μg/mL FITC-ferritin nanoparticles (H-Ft, Mutant A, B and C). Prior imaging, cells were washed twice with PBS to eliminate the unbound FITC-ferritins and then replaced in the incubation medium (suitable for confocal imaging purposes). The confocal laser-scanning microscope used was an Olympus FV10i platform equipped with a built-in incubator. Images

were acquired with a ×60/1.2NA water-immersion objective, LD lasers and filter sets for FITC. FITC and phase-contrast channels were acquired simultaneously.

## Data availability

All data supporting the findings of this study are available within this paper and from the corresponding authors. The cryo-EM maps of CD71/H-Ft complex at 5.5 Å and at 3.9 Å and coordinates generated and analyzed in the current study have been deposited in the Electron Microscopy Data Bank and in the Protein Data Bank under accession code EMD-0046 (PDB 6GSR) and EMD-0140 (PDB 6H5I), respectively. The source data underlying Fig. 3a, b and Supplementary Figs. 1a, 5 and 6 are provided as a Source Data file.

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

## Acknowledgements

We thank the H2CU, Honors Center of Italian University for support to B.V., C.T. and L. C.M., the Associazione Italiana Cristallografia (AIC) for support to L.C.M. and the Italian Academy for Advanced Studies at Columbia University for support to B.V. We are grateful to Dr. L. Shapiro (Columbia University) for the kind gift of the pα−H BiP vector, modified from the original pα−H optimized for protein secretion by Dr. D.J. Leahy (Texas University at Austin). We acknowledge Tong Wang and the ASRC Imaging Suite at the Graduate Center of CUNY (NY, USA) and to the European Synchrotron Radiation Facility for provision of microscope time on CM01 and we would like to thank E. Kandiah and G. Effantin for their assistance. The authors wish to thank the Imaging Facility at Center for Life Nano Science, Istituto Italiano di Tecnologia (IIT), for support and technical advices. We also thank D. Ben-Hail and F. Saaman (CUNY ASRC, NY, USA) for their support during initial screenings and cryo-EM data collection. This project has received funding from the European Union's Horizon 2020 research and

innovation program under the Marie Skłodowska-Curie grant agreement No. 823780 (B.V.). This research was supported by grants from the MIUR flagship Project "Nanomax" (AB) and Associazione Italiana per la Ricerca sul Cancro (AIRC) I.G. Grant 16776 (P.C.) and by CUNY startup funds to A.d.G. This work is dedicated to the memory of Professor Emilia Chiancone who pioneered the field of molecular biology of ferritins.

## Author contributions

A.B., A.d.G. and B.V. conceived the work and designed the project. B.V. performed cloning of CD71. L.C.M., C.T. and B.V. expressed and purified CD71. P.C., E.F. and M.P. designed mutations and expressed and purified human apo-ferritin and mutants. L.C.M. optimized the biochemical methods for complex formation and isolation. L.C.M. and C. T. prepared specimen for Cryo-EM and performed data collection. L.C.M., C.T., and A.d. G. performed data analysis, generating maps. L.C.M., C.T. and C.S. performed model building, refinement, and validation. C.T. and G.P. conceived and performed HeLa cells ferritin uptake, collected and analyzed confocal and FACS data. P.B. performed the FITC labeling of ferritins. A.A. collected, analyzed, and interpreted SPR-binding data. L.C.M., C.T., B.V. and A.B. interpreted data. L.C.M., C.T. and C.S. produced figures. L.C.M., C.T., F.M., A.B., B.V., A.d.G. wrote and finalized the manuscript.

## Additional information

**Competing interests:** The authors declare no competing interests.

