## [Peer Review File · Nature Communications]

Reviewers' comments:

Reviewer #1 (Remarks to the Author):

This manuscript, entitled "Cryo-EM structure of the human Ferritin-Transferrin Receptor 1 complex" by Linda Celeste Montemiglio et al. describes the interaction between ferritin H chain and TfR using a cryo-EM technology. The interaction site on TfR1 was distinct from where it associates with Tf, whereas it was partially common with the region where it associates some viruses. These results presented here are supported by some literatures which found that TfR1 associates with Tf and FtH using distinct recognition sites. I think these results deserve to be published as an article in Nature Communications. However, I have some questions and find something which should be described.

1. Validation of the interaction sites.

These authors found and verified some amino acid residues which are critical for the interaction between TfR1 and FtH, e.g., Q14, D15 and R22 in A-helix of FtH which are mutated in mutant A. The mutations reduced the amount of ferritin associated with TfR1 that caused the reduced incorporation into cells. All of these data are clear. I would like to see the result of gain of function experiments, such as mutation on the surface of human L chain ferritin which does not have these critical amino acids. Some of those residues on FtH which related to the association with TfR1 were common in both FtH and FtL. However, most of the amino acid residues selected for the mutation were specific to FtH. What if these residues would be introduced to FtL? Native FtL cannot bind to TfR (Line 127).

2. Method for the sample preparation.

Sample preparation of the complex is one of the critical points in these experiments. I would like to know more detailed method to get the TfR1-FtH complex. From line 167, the authors described the preparation method. The eluted complex was used for cryo-EM without any buffer exchange (Line 181)? Did the complex stay stable in imidazole containing buffer?

Minor points.

Line 117 FACS. What is FACS? Does it mean FITC?

Line 522 c. It should be f.

Reviewer #2 (Remarks to the Author):

Montemiglio et al presented cryoEM structure of the human ferritin-transferrin receptor 1 (CD71) complex. They identified and validated key structural contacts between the proteins. The findings are novel and important. The methodology is solid in general. I recommend publication of this manuscript if the authors can address my concerns as listed below:

1, The global resolution for Ferritin and the contact region with CD71 is reported to be at 3.9 Å. However, as the larger and more rigid volume of Ferritin dominates the calculation of GSFSC and global resolution, it is not very clear from the local resolution map what is the resolution at the contact region with CD71. The author should calculate model to map FSC curve focused mainly on the contact region between Ferritin and CD71, i.e. mask out most density in Ferritin and calculate the model to map FSC curve for the contact region.

2, It is a bit unusual that some FSC curves in extended data figure 2c and 2e have a dip and then go up before dropping again. Do the authors have explanations on what possibly causing the dip? Sometimes residual beam tilt during the cryoEM imaging process can cause a dip in the FSC curve. The recently released version 3.0 of cryoEM data processing software Relion can take such beam

tilt into account. I think the authors ought to explore this.

3, On page 11 lines 266-267, the description "sharpened using negative temperature B factors as calculated by RELION and cisTEM" is not accurate. cisTEM uses different strategy to sharpen a density map and it does not calculate a B factor. Instead the user needs to provide an empirical B-factor. The authors need to rephrase this.

Reviewer #3 (Remarks to the Author):

The authors have obtained for the first time a cryo-EM structure of a human ferritin-transferrin receptor1 complex. They find that ferritin binds to the same domain as the new world arenaviruses and the parasite Plasmodium vivax.

Major problem

1. The authors fail show whether the mutant forms of ferritin that do not bind to TfR1 fold and assemble properly. HPLC gel filtration of the wild type and mutant forms of ferritin or native blue gel non denaturing electrophoresis should be able to resolve this issue.

Minor issues;

2. Membrane bound receptors are classified as undergoing constitutive or triggered endocytosis. TfR1 is considered to undergo constitutive endocytosis whether or not Tf is bound. Epidermal growth factor receptor and insulin receptor fall into the triggered receptor category.

3. Line 74, the point maybe subtle but TfR1 is highly expressed in cancer cells.

4. A crystal structure of Tf-bound to the TfR2 ectodomain was published after the cryo-EM structure was done and has greater resolution. It is different from the Cryo-EM version. Eckenroth BE, Steere AN, Chasteen ND, Everse SJ, Mason AB. Proc Natl Acad Sci U S A. 2011 108:13089-94.

5. The major form of ferritin in the blood is L-ferritin. What significance do the authors attribute to binding H-ferritin and how it could alter arenavirus and Plasmodium vivax infections?

Reviewer #1

This manuscript, entitled “Cryo-EM structure of the human Ferritin-Transferrin Receptor 1 complex” by Linda Celeste Montemiglio et al. describes the interaction between ferritin H chain and TfR using a cryo-EM technology. The interaction site on TfR1 was distinct from where it associates with Tf, whereas it was partially common with the region where it associates some viruses. These results presented here are supported by some literatures which found that TfR1 associates with Tf and FtH using distinct recognition sites. I think these results deserve to be published as an article in Nature Communications. However, I have some questions and find something which should be described.

1. Validation of the interaction sites. These authors found and verified some amino acid residues which are critical for the interaction between TfR1 and FtH, e.g., Q14, D15 and R22 in A-helix of FtH which are mutated in mutant A. The mutations reduced the amount of ferritin associated with TfR1 that caused the reduced incorporation into cells. All of these data are clear. I would like to see the result of gain of function experiments, such as mutation on the surface of human L chain ferritin which does not have these critical amino acids. Some of those residues on FtH which related to the association with TfR1 were common in both FtH and FtL. However, most of the amino acid residues selected for the mutation were specific to FtH. What if these residues would be introduced to FtL? Native FtL cannot bind to TfR (Line 127).

Following the Reviewer’s request, we designed and produced a gain-of-function mutant (mutant D, new version of the manuscript) of the human L-chain ferritin (L-Ft), where residues S5, T14, S22, L81, D116, A119 and A123 were substituted with the corresponding amino acids of the H-chain ferritin (H-Ft), *i.e.* T5, Q14, R22, F81, E116, K119 and D123 (see **Supplementary Figure 4a**) involved in interaction with CD71. We obtained low purification yields for this mutant, since only a small fraction of the expressed protein is soluble (<5 %), while most of it forms inclusion bodies. However, we were able to obtain a pure sample of mutant D that retains the 24-mer assembly typical of human ferritin, as determined by HPLC analysis and shown in **Supplementary Figure 4b**.

We tested the binding affinity of mutant D to CD71 by SPR assay (**Supplementary Figure 6**), running the experiment in single cycle kinetics, as detailed in the Methods section, as the best way to compensate for the reduced amount of protein usable as analyte. In this experiment, H-chain and L-chain ferritins were used as positive and negative controls, respectively. As expected, the L-Ft proved ineffective in binding CD71 receptor, as also mentioned by the Reviewer, and consequently the SPR assay did not create the expected sawtooth kinetic profile. On the contrary, we found that the introduced mutations showed binding for L-Ft comparable to the one displayed by H-Ft. Indeed, mutant D shows a kinetic profile similar to the one of H-Ft, suggesting an affinity of the same order of magnitude for the CD71 receptor. This result shows unequivocally a gain-of-function of L-Ft mutant that became able to interact with the CD71 receptor. Accordingly, the following sentence has been added in the manuscript in the Results and Discussion section (page 5, lines 125-129): “Notably, Hum L-Ft, which is unable to bind CD71⁸, presents differences in seven positions

(S5, T14, S22, L81, D116, A119, A123) over the total contacts required for the recognition of the receptor (**Supplementary Figure 4a**). We found that L- to H-Ft mutations at these positions (mutant D) confers binding capability to CD71 with an affinity similar to the one observed for H-Ft (**Supplementary Figure 6**)". Therefore, in summary, thanks to the reviewer's request, we have produced a form of L-ferritin that is able to bind CD71 receptor, further validating the relevance of the identified residues in driving the CD71/H-Ft interaction.

2. Method for the sample preparation. Sample preparation of the complex is one of the critical points in these experiments. I would like to know more detailed method to get the TfR1-FtH complex. From line 167, the authors described the preparation method. The eluted complex was used for cryo-EM without any buffer exchange (Line 181)? Did the complex stay stable in imidazole containing buffer?

The revised version of the manuscript now includes a more detailed description of the sample preparation protocol. In brief, the solution containing the isolated CD71/H-Ft complex was applied to the grids immediately after elution from Talon resin (no more than 1 hour after), always keeping the sample on ice. No buffer exchange was performed. While we did not test the stability of the complex in those conditions over time, we observed that after one week the sample stored at 4°C did not present any spontaneous aggregate and no precipitation upon spinning could be observed.

In the revised version of the manuscript (page 8, lines 192-195) we specify that the sample was freshly prepared and immediately applied to the grids before plunge-freezing, without buffer exchange and concentration of the sample.

Minor points.

Q n. 1: *Line 117 FACS. What is FACS? Does it mean FITC?*

A. Line 114 of the revised manuscript: FACS stands for fluorescence activated cell sorting, and it is now written in full. FITC is the acronym for fluorescein-isothiocyanide, the fluorochrome used to label ferritin molecules used for FACS experiment, and this acronym is also given in full wording.

Q n. 2: *Line 522 c. It should be f.*

A. We apologize for this mislabeling. In the revised version of the manuscript the Table shown in **Figure 2f** was moved in Supplementary Materials (**Supplementary Table 3**) so as to follow the Nat Commun format where Tables cannot be shown in Figures.

Reviewer #2

Montemiglio et al presented cryoEM structure of the human ferritin-transferrin receptor 1 (CD71) complex. They identified and validated key structural contacts between the proteins. The findings are novel and important. The methodology is solid in general. I recommend publication of this manuscript if the authors can address my concerns as listed below:

1. The global resolution for Ferritin and the contact region with CD71 is reported to be at 3.9 Å. However, as the larger and more rigid volume of Ferritin dominates the calculation of GSFSC and global resolution, it is not very clear from the local resolution map what is the resolution at the contact region with CD71. The author should calculate model to map FSC curve focused mainly on the contact region between Ferritin and CD71, i.e. mask out most density in Ferritin and calculate the model to map FSC curve for the contact region.

Following the Reviewer's suggestion, we estimated the resolution on the interacting region between Ferritin and CD71 ectodomain by calculating the FSC curve obtained by masking the contact region only. This yielded a resolution of 3.9 Å for the interface, based on the gold-standard 0.143 FSC criterion. In the revised manuscript, the final curve is reported in **Supplementary Figure 2g** and the following sentence has been added in Methods (page 11, lines 278-280): "To estimate the resolution at the CD71/H-Ft interacting region we applied a spherical mask, created with Chimera⁴², only including this portion. The resolution obtained is 3.9 Å based on the gold-standard 0.143 FSC criterion (**Supplementary Figure 2g**)".

2. It is a bit unusual that some FSC curves in extended data figure 2c and 2e have a dip and then go up before dropping again. Do the authors have explanations on what possibly causing the dip? Sometimes residual beam tilt during the cryoEM imaging process can cause a dip in the FSC curve. The recently released version 3.0 of cryoEM data processing software Relion can take such beam tilt into account. I think the authors ought to explore this.

We thank the Reviewer for the point raised. We recalculated the FSC curves on the map at 4.2 Å using Relion 3.0 and on the map at 3.9 Å using cisTEM. It seems that we may have used a too tight mask for the postprocessing step in the former map and a too small mask radius in the latter map. As a result, we applied a wider and softer mask in Relion and a larger mask outer radius in cisTEM. In both cases the final trend of the FSC curves improved, and all curves now converge to zero at high spatial frequencies (new **Supplementary Figures 2c** and **2e**). However, given the slight left-shift of the intersection point with the FSC = 0.143 line, i.e. lower spatial frequencies, the estimated resolution changes from 4.2 to 4.4 Å, even if the overall quality of the yielded map was not affected. Conversely, the final resolution of the map at 3.9 Å did not change.

We also tried to estimate the beam tilt effect on the FSC curves using the specific command on CTF refinement implemented in Relion 3.0, but no improvement was obtained. Indeed, the program allows a reliable estimation of this parameter only for very high-resolution data sets, i.e. significantly beyond 3 Å resolution [ftp://ftp.mrc-lmb.cam.ac.uk/pub/scheres/relion30_tutorial.pdf]. Therefore, we cannot exclude an effect of

the beam tilt on the overall shape of the FSC curves which would not be corrected by using the option available in Relion 3.0. Accordingly, we modified the relative paragraphs in Methods.

3. On page 11 lines 266-267, the description “sharpened using negative temperature B factors as calculated by RELION and cisTEM” is not accurate. cisTEM uses different strategy to sharpen a density map and it does not calculate a B factor. Instead the user needs to provide an empirical B-factor. The authors need to rephrase this.

The sentence mentioned by Reviewer (page 11, line 281-282) has been rephrased as follows: “...sharpened using negative temperature B factors as estimated by RELION”.

Reviewer #3

The authors have obtained for the first time a cryo-EM structure of a human ferritin-transferrin receptor1 complex. They find that ferritin binds to the same domain as the new world arenaviruses and the parasite Plasmodium vivax.

Major problem

1. The authors fail show whether the mutant forms of ferritin that do not bind to TfR1 fold and assemble properly. HPLC gel filtration of the wild type and mutant forms of ferritin or native blue gel non denaturing electrophoresis should be able to resolve this issue.

Following the Reviewer's comment, we performed Size Exclusion Chromatography (SEC)-HPLC experiments for all ferritins examined in the paper. Resulting chromatograms of all the proteins analysed (now reported in **Supplementary Figure 4b**) show the presence of a main peak at retention times typical of the human ferritin wild type (28.2 min), thus proving evidence that all mutants retain the 24-mer assembly of the human wild type ferritins. In addition, a pre-peak at about 25 min is also present, although only in the H-chain constructs (including the wild type). This was previously described as due to the presence of cysteines (2 per subunit, 48 for 24-mer) on the surface of the H-chain ferritins that are absent in the L-chain proteins (Niitsu Y, Listowsky I., *Biochemistry*, 1973).

Minor issues:

Q n. 2: *Membrane bound receptors are classified as undergoing constitutive or triggered endocytosis. TfR1 is considered to undergo constitutive endocytosis whether or not Tf is bound. Epidermal growth factor receptor and insulin receptor fall into the triggered receptor category.*

A. We modified the abstract and rephrased two sentences in Introduction session of the revised manuscript:

- i) Page 2, lines 40-42. "Iron uptake is mediated by the internalization of the transferrin-iron complex through receptor-mediated **constitutive** endocytosis *via* a clathrin-dependent pathway";
- ii) Page 2, lines 46-48. "Viral systems recognize epitopes on the host-encoded CD71 receptor through their surface spike glycoproteins, **allowing** the internalization of the complex".

Q n. 3: *Line 74, the point maybe subtle but TfR1 is highly expressed in cancer cells.*

A. Corrected.

Q n. 4: *A crystal structure of Tf-bound to the TfR2 ectodomain was published after the cryo-EM structure was done and has greater resolution. It is different from the Cryo-EM version. Eckenroth BE, Steere AN, Chasteen ND, Everse SJ, Mason AB. Proc Natl Acad Sci U S A. 2011 108:13089-94.*

A. We thank the Reviewer for this observation. We used the more accurate information on the aminoacids of CD71 involved in interaction with Tf in **Figure 1a** and we refer to the above

mentioned manuscript throughout the revised version of the paper. However, in Figure 1b we kept the cryo-EM model at lower resolution reported for the whole Tf-TfR structure (pdb 1SUV, Ref. n. 14 in the manuscript) since the crystallographic model lacks one of the C-lobe subunit.

Q n. 5: *The major form of ferritin in the blood is L-ferritin. What significance do the authors attribute to binding H-ferritin and how it could alter arenavirus and Plasmodium vivax infections?*

A. Each individual ferritin assembly is composed by different ratios of H and L subunits depending on the tissue and the organism [Harrison PM, Arosio P, *Biochim Biophys Acta*, 1996]. Therefore, even if serum ferritins are composed mostly by L-subunits, the H-chain is always present at least in one copy in the 24-mer assembly (Ghosh S, Hevi S, Chuck SL, *Blood*, 2004). Indeed, our results show that even a single subunit of H-ferritin over 24 can be recognized by CD71 and this might be sufficient to favor cellular internalization of the mixed H/L chain 24-mer ferritin through this route of access. This scenario is in agreement with the results reported by Zhang L *et al.* (H-Chain Ferritin: A Natural Nuclei Targeting and Bioactive Delivery Nanovector. *Adv Healthc Mater.* 2015, 4(9):1305-10), showing that having a small H-chain fraction of the 24-mer is sufficient to drive CD71-based internalization of L/H mixed apoferritin. Nevertheless, the physiological function of serum ferritin still remains unclear [Wang W, Knovich MA, Coffman LG, Torti FM, Torti SV, *Biochim Biophys Acta*, 2010]. The structural evidence that we provide of a not-competitive binding to CD71 between transferrin and ferritin supports the hypothesis that the existence of the two iron-transporter proteins may serve as redundant mechanism for the uptake of iron by cells through the same portal [Ref. n. 8 in the manuscript].

The following sentence has been added in Results and Discussion (page 6, lines 131-134): “Our results show that even a single subunit of H-chain over 24 can be recognized by CD71 and this might be sufficient to favor the cellular internalization of the mixed H/L-chain 24-mer ferritin through this route of access, even though in serum the L-chain is prevalent²⁹.”

Regarding the effect of H-Ft binding to CD71, we propose that it represents the physiological competition with arenavirus and *P. vivax* since H-Ft interacts with residues on the CD71 apical domain, essential for pathogen internalization. Therefore, following the work of Radoshitzky *et al.* where an anti-CD71 apical domain was proved to inhibit infection of several types of arenavirus [Ref. n. 7 in the manuscript], the structural information provided by our work might be exploited to produce ferritin-like peptides specifically targeting “common residues” of the apical domain to compete with virus glycoproteins and parasite PvRBP2b, thus reducing their internalization and inhibiting their infectious capability.

REVIEWERS' COMMENTS:

Reviewer #1 (Remarks to the Author):

In the revised manuscript, the authors addressed all the questions I suggested. This manuscript sheds light on the mechanism in formation of the complex of TfR1 and ferritin that is desired information not only for researchers in iron metabolism.